# Human Endogenous Retroviruses (HERVs): Shaping the Innate Immune Response in Cancers

**DOI:** 10.3390/cancers12030610

**Published:** 2020-03-06

**Authors:** Vincent Alcazer, Paola Bonaventura, Stephane Depil

**Affiliations:** 1Cancer Research Center of Lyon, 69008 Lyon, France; 2Department of Clinical Hematology, Centre Hospitalier Lyon Sud, Hospices Civils de Lyon, 69310 Pierre-Bénite, France; 3Centre Léon Bérard, 69008 Lyon, France; 4Université Claude Bernard Lyon 1, 69008 Lyon, France; 5ErVaccine Technologies, 69008 Lyon, France

**Keywords:** human endogenous retroviruses, innate immunity, cancer

## Abstract

Human Endogenous Retroviruses (HERVs) are accounting for 8% of the human genome. These sequences are remnants from ancient germline infections by exogenous retroviruses. After million years of evolution and multiple integrations, HERVs have acquired many damages rendering them defective. At steady state, HERVs are mostly localized in the heterochromatin and silenced by methylation. Multiple conditions have been described to induce their reactivation, including auto-immune diseases and cancers. HERVs re-expression leads to RNA (simple and double-stranded) and DNA production (by reverse transcription), modulating the innate immune response. Some studies also argue for a role of HERVs in shaping the evolution of innate immunity, notably in the development of the interferon response. However, their exact role in the innate immune response, particularly in cancer, remains to be defined. In this review, we see how HERVs could be key-players in mounting an antitumor immune response. After a brief introduction on HERVs characteristics and biology, we review the different mechanisms by which HERVs can interact with the immune system, with a focus on the innate response. We then discuss the potential impact of HERVs expression on the innate immune response in cancer.

## 1. Introduction

About half of the human genome is composed of transposable elements (TEs) [1,2]. Initially considered as “junk DNA”, these mobile sequences have been shown to provide a major source of structural variation in the genome with an impact on oncogenesis [1]. Among them, Human Endogenous Retroviruses (HERVs) represent 8% of the human genome. These DNA sequences of retroviral origin are the legacy of ancient germ line infections by exogenous retroviruses, persisting as integrated retroviruses. HERVs complete structure is the same as that of exogenous retroviruses, consisting of four genes (*gag*, *pro*, *pol* and *env*) flanked by two long-terminal repeats (LTRs) (Figure 1A) [3]. These LTRs function as promoters for HERV expression, have strong RNA regulatory sequences and contain transcription factor binding sites. Successive recombination between the two LTR led to internal deletion in the vast majority of HERVs, leading to the excision of most of the coding regions [2]. The hundred thousand copies of HERVs present in the human genome is thus mainly constituted of solitary LTR, only few thousands among the most recent integrated ones (such as the HML family) keeping partially conserved open reading frame (ORF) (Figure 1B,C). It is important to note that humans differ from other mammals in this point, and particularly from mice which, unlike humans, possess complete replication-competent ERVs sequences [4].

HERVs are mainly localized in heterochromatin and repressed by epigenetic silencing at steady state [5]. Only few physiological functions have been linked to HERVs expression so far. The best known physiological example are Syncitins 1 and 2, two HERV-W/HERV-FRD derived envelope proteins expressed in the placenta playing a role in trophoblast formation and foeto-maternal tolerance [6]. Different studies also suggest a role for HERVs in the complex human brain development: In human neural progenitor cells, HERVs act as a docking platform for the epigenetic co-repressor protein TRIM28, inducing a local state of heterochromatin around these HERVs [7]. It has been shown that the transfer from motoneuron to muscles of Arc/Arg3.1, a crucial protein for synaptic plasticity, is regulated by a Gag-like protein contained in its 3’ UTR region [8]. Conversely, increased HERVs transcription and activity has been described in several pathological conditions, including auto-immune diseases and cancer [1,4]. Indeed, the loss of CpG methylation in tumors seems to preferentially affect retrotransposon elements and particularly HERVs [4].

Due to their retrotransposition ability, HERVs can be involved in oncogenesis in several ways such as insertional mutagenesis or chromosomal instability [1]. LTR-induced oncogene transcription has also been reported recently [9]. Furthermore, HERV-derived proteins can exert direct oncogenic activity, as it is the case for instance with Np9, an HERV-K derived protein that can activate Wnt/β-catenin, Ras-ERK, cMyc/AKT and Notch1 signaling pathways in chronic lymphocytic leukemia [10]. 

HERVs may have different impacts on the immune response. On one hand, the transmembrane subunit of the Envelope (Env) protein from different HERVs contains a conserved immunosuppressive domain (ISD) [6]. Of note, the mechanisms and targets of Env-mediated immunosuppression are not fully defined, and its role in cancer immune escape is not clearly established. On the other hand, HERVs’ replication intermediates can stimulate innate sensors promoting antitumoral immune response by triggering a type I and III Interferon (IFN) response [11,12]. HERVs expression is thus a double-edged sword for antitumor immunity, favoring either immune escape or immune activation.

In this review, we will present the main data linking HERVs expression with the innate immune response in cancer. After a brief review of the mechanisms regulating HERVs expression, we will discuss the interactions between HERVs and the innate immune response, before considering the implication of HERVs expression in the antitumor immune response.

## 2. HERVs Expression: Interaction with the Innate Immune System

### 2.1. HERVs Are Silenced by Epigenetic Mechanisms at Steady State

Epigenetics plays a key role in embryonic development, allowing a fine regulation of cell fate and pluripotency and contributing to HERVs repression [5]. Histone H3 trimethylation, at the lysine 9 position (H3K9me3) by the histone-lysine N-methyltransferase SETDB1, is involved in early HERVs’ silencing during embryogenesis, while DNA methylation is thought to be the main mechanism in differentiated cells. As more than 90% of methylated CpGs are located in part of the genome harboring TEs, some authors even suggested that cytosine methylation was developed as a defense mechanism against TEs spreading in human [11,13]. The exact role of each mechanism is, however, still a matter of debate, different studies suggesting a more generalized role for SETDB1 in differentiated cells such as lymphocytes [14,15]. In mice, SETDB1 was shown to ensure Th2 cell stability by repressing ERVs that control the Th1 gene network [15], and also to repress the expression of FcγR-IIb in T cells, an inhibitory Fc receptor regulating the signaling through the TCR complex [16]. Accordingly, SETDB1 knock out induces an increased phosphorylation of ZAP70 in response to CD3 agonism, resulting in an increased cell activation and death at early thymic selection stage. In B cells, loss of SETDB1 induces a significant expression of HERVs, incompatible with B cell development and survival [14,17]. 

TRIM28 may also play an important role as shown in human neural progenitor cells by inducing a local heterochromatin state to suppress HERVs expression [7]. Derepression of this local network by knockdown of TRIM28 induces the transcription of both HERVs and their surrounding genes, providing another layer of epigenetic control in these cells. Another key player is the retinoblastoma protein (pRB) which recruits EZH2 to establish trimethylation of the 27th lysin of histone H3 (H3K27me3, a marker of heterochromatin) on tandem and interspersed genomic repeats in somatic cells [18]. Multiple factors are thus participating in the control of HERVs expression through epigenetic modifications, and there is still a need to better understand and identify all the actors involved in this regulation.

### 2.2. HERVs Expression Is Modulated by the Immune Response, Microbiota and Viral Infections

Multiple immune stimuli have been shown to induce ERVs expression. First evidences of Murine Leukemia Virus (MLVs) induction following murine B cells activation were reported in the 80’s by Stoye J. and Moroni C [19]. The authors showed that both B cell mitogens (lipopolysaccharide (LPS) and lipoprotein (LP)) and 5-bromo-2’-deoxyuridine (BrdU) were able to induce ERVs expression in mice, and that their combination was synergistic, BrdU preventing the methylation of newly replicated DNA. More recently, Young G. and Kassiotis G. assessed ERVs expression by microarray in murine and human immune cells under different stimulatory conditions [20]. ERVs assessment in gut tissues from wild type or MYD88 knock out mice breaded in two different conditions (specific pathogen-free versus germ-free) revealed that expression of certain ERVs was dependent on the presence of the gut microbiota. Consequently, ERVs expression was largely decreased in MYD88-/- mice, supporting the role of the microbiota in inducing a basal ERVs expression in mice’s gut through MYD88 signaling. Still in mice, deficiency in Toll-like Receptors (TLR)-3 and -9 (recognizing double stranded (ds) RNA) or in TLR-7 (recognizing single stranded (ss) RNA) was shown to allow endogenous MLVs reactivation [21].

In human, analysis of blood samples from HIV-1 infected patients show an increase in HERV-K RNA expression [22]. HERV-K protein production is also increased after HIV-1 infection of CD4 T cell lines [23]. Similar results were found in gut biopsies from ulcerative colitis patients compared to healthy individuals, confirming that not only external pathogens but also commensal bacteria can modulate HERVs expression [20]. Peripheral blood mononuclear cells (PBMCs) treated by type I IFN express high levels of Alu elements, and IFN-α stimulation following a viral infection is known to induce transcription of certain HERVs [24,25]. In this latter case, it was clearly shown that the expression of HERV-K18 Env-derived superantigen IDDMK1,222, which is known to induce the activation of autoreactive T cells associated with type I diabetes development, is strongly induced by IFN-α [25]. Type II IFN can also promote the expression of a particular class of HERVs, located in the 3’ untranslated region of specific genes enriched for STAT1 and EZH2 regulation motifs [26]. IFN-γ exposure induces the bidirectional transcription from these sequences, leading to dsRNA generation with 3’ antisense retroviral coding sequences (labelled SPARCS which correspond to antisense ERVs) transcription.

Other mechanisms can participate to ERVs transcription in case of viral infection. Small ubiquitin-like modifiers (SUMO) are a family of proteins involved in the regulation of multiple cellular proteins and pathways by inducing post-translational modifications [27]. SUMOylation has been shown to be a key process during the host response to viral infection by suppressing the polyubiquitination and degradation of PRRs such as RIG-1 and MDA-5 [28]. There is also evidence for TRIM28 SUMOylation, but not phosphorylation, to be implicated in ERVs repression [29]. Recently, it has been shown that influenza virus infections induce loss of a SUMO-modified form of TRIM28, leading to a derepression of ERVs [30]. The resulting dsRNA-mediated type I IFN response enhances the host antiviral response, illustrating how an aberrant transcription of self-RNA derived from ERVs can be sensed as non-self by PRRs during an infection. Nevertheless, the IFN-antagonist protein NS1 produced by the wild-type influenza virus can oligomerize around dsRNAs to mitigate the antiviral effects of ERVs transcription [30]. 

Adaptive immunity could also play a role in the control of ERVs expression. Antibody-deficient mice present the spontaneous expression of fully infectious MLVs copies, leading to the development of lymphomas [31]. Evidences of specific CD8^+^ and CD4^+^ T cells directed against HERV-derived peptides have also been reported, as discussed later.

### 2.3. Role of PRRs, TLR Signaling and B Cell Response in the Control of ERVs 

While studies have provided a better understanding of the different mechanisms underlying HERVs expression, the link between HERVs and the immune system still remains poorly defined, with most evidences coming from exogenous retroviruses or mice models ([2] for review). As discussed above, TLR deficiency can lead to ERVs reactivation in mice [21]. Yu et al. showed that while TLR-7 deficiency alone leads to spontaneous viremia, additional TLR-3 and -9 deficiencies lead to the development of acute lymphoblastic T cell lymphoma. Interestingly, TLR-7 deficient mice expressed ERVs but lacked ERVs-specific antibody response, confirming a previous study showing that TLR-7 is essential for initiating a B cell response directed against ERVs, and that CD4^+^ T cell responses are also attenuated in the absence of TLR-7, whereas CD8^+^ T cell responses are conserved [32]. In this latter study, the authors showed that intrinsic TLR signaling in B cells is mediating the response, while TLR signaling in DCs is less important. [33]. Various levels of antibody specificities and efficiencies in TLR-3 and 9 simple or double deficient mice also suggest a modulatory role for these two TLRs in ERV-specific antibody response. Of note, both studies are describing responses against a fully infectious murine ERV, the exogenous Friend-MuLV retrovirus, and may not be extrapolated to human ERVs, since most of them are defective. 

Altogether these data illustrate how the immune system modulates and controls ERVs expression, leading to malignancy in case of immune deficiency. Recently, it has been shown that interferon-γ-inducible protein 16 (IFI16), an AIM2-like receptor for DNA, is able to sense HERV-K-derived ssDNA in somatic tissues, while being absent in stem cells [34]. It seems likely that different mechanisms evolved to provide a better control of an aberrant transcription of these sequences in somatic tissues while allowing expression in stem cells where they may contribute to cell identity. 

### 2.4. HERVs as Network Regulator: the Example of the IFN-γ Network 

With their ability to move and replicate within the genome, TEs have provided a major source of gene regulation. Indeed, once inserted, TEs can provide an alternative promoter sequence and introduce an alternative transcription start site, interrupt a pre-existing promoter, introduce a transcription factor binding site, drive an antisense transcription or silence a whole region by inducing a local state of heterochromatin [35]. TEs can also act post-transcriptionally by providing an alternative polyadenylation site, an alternative splicing or a target for micro RNA. Among the multiple examples of gene network regulated by TEs, it is now clear that HERVs participated in the shaping and the development of the IFN-γ network [36]. By providing already functional cis-regulatory elements to surrounding genes, TEs insertion promotes emergence of complex gene regulatory networks that would take much longer time to develop by independent de novo mutations [37]. Chuong et al. thus found 27 families of TEs enriched within IFN-γ-responsive cis regulatory elements, among which 20 were LTRs from the HERV family with STAT1 binding motifs [36]. The authors also showed that the only STAT1 binding site within the 50 kb of the Absent in Melanoma 2 gene (AIM2, an IFN-stimulated gene encoding a sensor of foreign cytosolic DNA promoting the assembly of the inflammasome) is located in a LTR sequence, derepressed by H3K27 acetylation after IFN-γ exposure. Suppression of this LTR element by Crispr-Cas9 completely abrogated AIM2 expression upon IFN-γ treatment. Activated capase-1 levels, a surrogate marker of pyroptotic cell death, were also markedly reduced after vaccinia-virus infection, demonstrating that this LTR element is indispensable for the inflammatory response to infection. In line with this, an impaired activity of IFN-stimulated genes is observed in cell lines deleted of these HERV sequences [36]. HERVs may thus have been key actors in the development and the shaping of our immune system, by providing ready-to-use IFN-γ inducible enhancers.

## 3. HERVs as Modulators of the Innate Immune Response

### 3.1. Activation of Innate Immunity by HERV Nucleic Acids

Pattern recognition receptors (PRRs) are key receptors of the innate immune system. Their ability to directly recognize conserved pathogen-associated molecular patterns (PAMPs) allow them to quickly trigger the innate immune response, acting in first line after an infection. Among them, two main families are specialized in the recognition of nucleic acids: endosomal PRRs (the TLR family), and cytosolic PRRs (including the RIG-I-Like Receptors (RLRs), the NOD-like receptors (NLRs), the C-type lectin receptors (CLRs) families and analogous DNA sensing receptors including oligoadenylate synthase (OAS) proteins and cyclic guanosine monophosphate–adenosine monophosphate (cGAMP) synthase (cGAS)) [38]. Once engaged, PRRs trigger different signaling cascades leading to inflammatory cytokines production and converging to type I IFN production. The pathways involved depends on the type of PRR. Schematically, TLRs signal through MYD88 or TRIF, while RIG-I (which recognizes short dsRNA and RNA with 5’ triphosphate ends) and MDA5 (which recognizes long dsRNA) both signal through MAVS, and DNA sensors signal through STING [38]. All together these effectors give rise to the innate immune response and the “danger signal” required for inducing a specific adaptive response [39].

The recognition of nucleic acids is at the time a very efficient strategy to fight viral infections and paradoxically a risky strategy due to the possibility of recognizing self-nucleic acids, promoting autoimmunity. With their ability to produce both RNA and DNA intermediates (by reverse-transcription), ERVs are constituting perfect candidates for stimulating PRRs, as complementary sense and anti-sense ERVs transcripts are forming both ssRNA and dsRNA. SsRNA can be sensed by TLR-7 and 8 resulting in IFN-α secretion by stimulated dendritic cells (DCs) and macrophages, as shown with HIV [40]. DsRNA, which are not found in normal cells, could be one of the most immunogenic nucleic acid PAMPs. HERV dsRNA can be recognized by TLR-3, RIG-I and MDA5, the two latters signaling through MAVS to induce a type I IFN response [11].

Once retrotranscribed into DNA, retroviruses can be sensed by cGAS to produce cGAMP, which binds and activates STING to trigger a type I IFN response trough nuclear factor-kappa B (NF-kB) and IFN regulatory factor 3 (IRF3) activation [41]. DsDNA could also be sensed by DNA-dependent activator of IFN-regulatory factors (DAI), and DNA:RNA hybrids by TLR-9, even if this has not been demonstrated for the particular case of HERVs [42,43]. 

Regulation mechanisms exist, such as 3′ repair exonuclease 1 (TREX1), a 3′→5′ DNA exonuclease that degrades retroviral DNA, thus preventing their accumulation [44]. TREX1 knock down increases type I IFN production after HIV infection in mucosal cells [45]. Albeit performed in mice, these experiments suggest that similar mechanisms could exist for modulating immune activation from nucleic acid sensing.

### 3.2. Activation of Innate Immunity by HERV- Derived Proteins

Not only nucleic acids but also proteins patterns can stimulate the innate immune response. Transmembrane TLR-2 and 4 are able to recognize retroviral envelope glycoproteins (at least of exogenous origin as shown for HIV-1), triggering a pro-inflammatory response through NF-kB activation [46]. Specific data regarding HERVs are still lacking to further extrapolate these results. Envelope subunit of multiple sclerosis-associated retroviral element (MSRV) from the HERV-W family was shown to specifically activate immune cells through CD14 and TLR-4, leading to production of pro-inflammatory cytokines such as IL-1β, IL-6 and TNF-α. It can activate DCs to promote the development of a Th1-response [47]. This could be an important immune activation pathway in the pathogenesis of multiple sclerosis, as suggested by a multiple sclerosis mouse model where HERV-W Env (Syncitin) overexpression in astrocytes led to neuroinflammation and oligodendrocytes’ death [48]. 

### 3.3. Suppression of the Immune Response

Most exogenous retroviruses have been described to suppress the host’s immune system to maintain the infection [49]. Regarding ERVs, this effect has been co-opted for the foeto-maternal tolerance, promoted by the expression in the placenta of Syncitin 2, a HERV-FRD-derived Env protein [50]. This shared property between exogenous and endogenous retroviruses led to the identification of an immunosuppressive domain (ISD) in the transmembrane unit of the Env protein (reviewed in [51]). Exposure of human PBMCs to ISD-derived peptides from different retroviruses, including HIV and HERV-K, increases the expression of different cytokines, including IL-10, IL-6, IL-8, RANTES, MCP-1, MCP-2, TNF-α, MIP-1α, MIP-1β, MIP-3, IL-1β and Gro(α,β,γ), and decreases the expression of IL-2 and CXCL9 [52,53]. Although all studies converge so far to a direct effect on cellular immune effectors, the exact impact of ISD remains unclear, and no interaction with the innate immune response has been clearly highlighted. An interesting study was performed in zebra-fish embryos, which have only innate immunity during the first month of development [54]. Embryos injected with HERV-K-containing extra-cellular vesicles derived from two colorectal adenocarcinoma cell lines treated with decitabine, showed a reduced expression of IL-1β and myeloperoxidase. This effect was however modest and needs to be investigated in other models. Of note, it has been also suggested that HERV-derived Env proteins could target T cell activation indirectly by modulating the stimulatory activity of dendritic cells [55].

## 4. HERVs as Innate “Adjuvants” in Cancer

### 4.1. Turning Cancerous Cells into Virus-Infected Cells

While the effects of 5-Azacytidine (AZA) in promoting ERVs expression had been described since 1984, the link with its anti-tumoral effect in human was only recently established [11,56]. Studies first showed the link between the use of AZA and the modulation of multiple immune pathways, including upregulation of immunomodulatory pathways and cancer testis antigens [57]. The classification of primary human tumors based on the union of these pathways confirmed the enrichment of this AZA-induced Immune Genes (AIM) after AZA therapy in breast and colorectal cancer trials, suggesting that patients with low AIM signature at diagnosis might benefit most from receiving epigenetic therapy prior to immunotherapy.

Chiappinelli K. et al. showed that AZA-induced response was mediated by cytosolic sensing of dsRNA via TLR-3 following expression of demethylated HERVs, leading to a type I IFN production and apoptosis of tumor cells. This activation could stimulate anti-tumor immunity by activating innate lymphoid cells and promoting the infiltration and clonal expansion of tumor-specific cytotoxic T cells (“viral mimicry mechanism”). The induction of HERVs expression could also lead to the exposure of several HERV-derived tumor-associated antigens, leading to new targets on the tumor cells [58]. These results were confirmed by Roulois D. et al. with the use of a slightly different DNA methyltransferase inhibitor (DNMTi), 5-aza-2′-deoxycytidine (AZA-CdR) in enriched sphere cultures derived from human colorectal cancer [12]. A low dose of 5-AZA-CdR also induced type I IFN and reduced the frequency of colorectal cancer initiating cells (CICs) independently of the cancer’s methylation profile. Knockdown of MDA5, MAVS, or IRF7 was sufficient to render these cells insensitive to 5-AZA-CdR treatment, in contrast with RIG1 knockdown, suggesting a major role for MDA5 sensor in 5-AZA-CdR efficiency. Interestingly, while 5-AZA-CdR also induced the expression of IFN-responsive genes, the MDA5-mediated apoptosis of tumor cells seemed to be independent of the IFN response, as already suggested in melanoma [59]. 

Other groups found similar results in different cancer types. In myelodysplastic syndromes, decrease in DNA methylation following AZA treatment was shown to be enriched in heterochromatin zones and not in coding genes, suggesting that HERVs activation is one of the mechanisms contributing to AZA clinical effects in these diseases [60]. High-grade glioblastoma, which exhibits repressive chromatin marks (by loss of H3K27 trimethylation) at some sites, contains reciprocal active chromatin marks (by gain of H3K27 acetylation) enriched at repeated elements sites, leading to their increased expression [61]. This acetylation can in fact be seen as a therapeutic vulnerability, rendering TEs and HERVs more susceptible to activation by epigenetic therapies such as AZA or histone deacetylase inhibitors (HDACi).

HERVs expression was also shown to overcome the resistance to anti-PD1 therapy in a melanoma mouse model: the ablation of the histone H3K4 demethylase LSD1 (KDM1A) enhances tumor immunogenicity and T cell infiltration by inducing dsRNA production through HERVs expression, leading to a type I IFN response [62]. This could be a way to turn a so-called “cold” into a “hot” tumor.

In a different approach, Cyclin-dependent kinases 4 and 6 inhibitors (CDK4/6is) have been shown to increase the anti-tumoral immune response by activating the expression of HERVs in tumor cells [63]. By inhibiting the phosphorylation of the pRB, they reduce DNMTs activity, promoting the expression of HERVs and other genes regulating immune functions. The increased intracellular levels of dsRNA stimulate the production of type III IFN, enhancing tumor antigen presentation and promoting a specific cytotoxic T cell response. CDK4/6is also lead to a selective suppression of regulatory T cell proliferation, potentially related to the higher expression of pRB by these latter. As with DNMTis, CDK4/6is might enhance the efficacy of immunotherapy such as immune checkpoints blockade. 

Altogether these results explain the mechanism behind the anti-tumoral effects of DNMTis, characterized by a delayed response time and paradoxically an absence of predictive methylation markers. By mimicking viral infection into the tumor cell through HERVs expression, epigenetic therapies could thus represent one approach to turn “cold” into “hot” tumors by promoting immune-cells infiltration and antigen presentation through type I IFN production while triggering cancerous-cell apoptosis and providing new targets for adaptive response [58,64]. All these effects put DNMTis as privileged candidates for association with immune therapies.

### 4.2. Impact of HERVs Expression in the Response to Immune Checkpoint Inhibitors 

By activating both innate and adaptive immune response, HERVs may represent key factors in the response to immunotherapy. HERV expression has recently been associated with response to anti-PD1 in clear cell renal cell carcinoma [65]. The “viral defense” gene expression signature associated with HERVs expression is also predictive of immune checkpoint responses in melanoma patients, and DNMTis treatment sensitize to anti-CTLA-4 therapy in a mouse melanoma model [11]. In the same way, Heidegger S. and Wintges A. et al. showed that RIG-I activation is indispensable for responsiveness to CTLA-4 checkpoint blockade [66]. 

HERVs could thus be involved in the response to checkpoint blockade in tumors characterized by a low mutational burden. It has recently been shown that in rhabdoid tumors, a pediatric cancer exhibiting one of the lowest mutational burden, the driver mutation characterized by a biallelic loss of SMARCB1 (a core member of a chromatin remodeler complex) triggers the expression of HERVs, explaining thus the unusual immune infiltration of these tumors, which is driven by the subsequent type III IFN response [67].

HERVs expression may also modulate the antitumor immune response once the latter has been initiated. For instance, SPARCS expression following IFN-γ exposure could be a positive feedback mechanism fostering the antitumoral immune response [26].

## 5. Conclusions

The integration of ERVs in the genome represents a good example of biological exaptation. ERVs have been co-opted by the host during evolution to modulate key functions and complex gene regulatory networks (such as IFN-γ). ERVs have also shaped the immune system, which, in turn, may control their expression. Since ERVs represent major factors of activation of the innate immune response, the induction of their expression in tumor cells is of great therapeutic interest. Indeed, many epigenetic drugs may act through ERVs derepression. This could represent a key mechanism to transform cold tumors into hot ones, justifying the rationale of combination of epigenetic drugs with immune checkpoint inhibitors.

With the potential to stimulate both the innate response through DNA/RNA sensing and the adaptive immune response through antigens recognition [68,69,70], ERVs provide unique and promising targets for cancer immunotherapy (Figure 2).

## Figures and Tables

**Figure 1 cancers-12-00610-f001:**
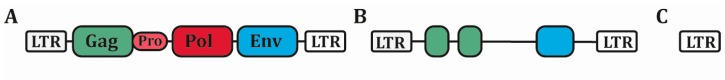
Structure of HERVs. (**A**) The complete proviral sequence of HERVs is composed of four protein-coding sequences (Gag, Pro, Pol and Env) surrounded by two long-terminal repeat (LTR) promoter sequences. (**B**) Most HERVs have acquired many damages on their ORFs, rendering them defective. Still, protein-coding sequences remain in some HERVs. (**C**) The vast majority of HERVs is now composed of solitary LTR. HERVs: Human Endogenous Retroviruses, LTR: Long-terminal Repeat, ORFs: Open-reading frames.

**Figure 2 cancers-12-00610-f002:**
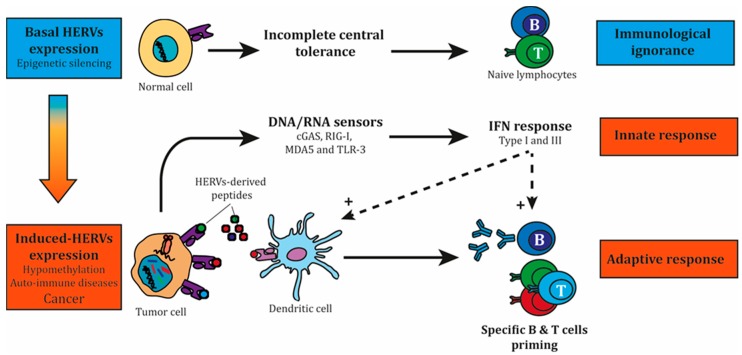
HERVs as relevant and unique targets for cancer immunotherapy. At steady state, most HERVs are not expressed, mainly silenced by epigenetic mechanisms. Basal HERV expression can be found in some tissues and in the thymus, with an incomplete central tolerance. Different circumstances such as auto-immune diseases or cancer can foster HERVs expression, leading to the production of both HERVs-derived nucleic acids and proteins. These intermediates can promote an innate immune response by inducing type I and III IFN production through DNA/RNA sensors, and also an adaptive immune response through B and T cell epitopes. IFN: Interferon.

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
