# Peer review of "Human Endogenous Retroviruses (HERVs): Shaping the Innate Immune Response in Cancers"

_cancers, 2020, doi:10.3390/cancers12030610_

Round 1

Reviewer 1 Report

The review by Alazer and colleague described the current scenario of the relationship between Human Endogenous Retroviruses and host innate immune response, with a specific focus on tumors.

Also basing on the poorness of original researches and review works on this topic, the manuscript deserves to be published. 

It is well written , clear, and the figure 2 well summarizes the formulated hypothesis.

The only minor issue regards the references, which should be updated. in the last few months some interesting papers were published, and they should be ddiscussed and added. here below , just some examples:  

1: Balestrieri E, Cipriani C, Matteucci C, Benvenuto A, Coniglio A, Argaw-Denboba A, Toschi N, Bucci I, Miele MT, Grelli S, Curatolo P, Sinibaldi-Vallebona P.
Children With Autism Spectrum Disorder and Their Mothers Share Abnormal
Expression of Selected Endogenous Retroviruses Families and Cytokines. Front Immunol. 2019 Sep 26;10:2244. doi: 10.3389/fimmu.2019.02244. 

2: Schmidt N, Domingues P, Golebiowski F, Patzina C, Tatham MH, Hay RT, Hale BG.  An influenza virus-triggered SUMO switch orchestrates co-opted endogenous retroviruses to stimulate host antiviral immunity. Proc Natl Acad Sci U S A. 2019 Aug 27;116(35):17399-17408. 

3: Ferrari L, Cafora M, Rota F, Hoxha M, Iodice S, Tarantini L, Dolci M, Delbue
S, Pistocchi A, Bollati V. Extracellular Vesicles Released by Colorectal Cancer
Cell Lines Modulate Innate Immune Response in Zebrafish Model: The Possible Role  of Human Endogenous Retroviruses. Int J Mol Sci. 2019 Jul 26;20(15). pii: E3669. 

4: Hurst TP, Aswad A, Karamitros T, Katzourakis A, Smith AL, Magiorkinis G.
Interferon-Inducible Protein 16 (IFI16) Has a Broad-Spectrum Binding Ability
Against ssDNA Targets: An Evolutionary Hypothesis for Antiretroviral Checkpoint. Front Microbiol. 2019 Jul 4;10:1426. doi:10.3389/fmicb.2019.01426. 

Author Response

We would like to thank the reviewer 1 for these pertinent references and his global appreciation of our work. Multiple sections have been added:

Section 2.2 has been completed to introduce ref 2 (pages 3-4, lines 135-146):

Other mechanisms can participate to ERVs transcription in case of infection. Small ubiquitin-like modifiers (SUMO) are a family of proteins involved in the regulation of multiple cellular proteins and pathways by inducing post-translational modifications [27]. SUMOylation has thus been shown to be a key process during the host response to viral infection by suppressing the polyubiquitination and degradation of PRRs such as RIG-1 and MDA-5 for example [28]. There is also evidences for TRIM28 SUMOylation, but not phosphorylation, to be implicated in ERVs repression [29]. Recently, it has been shown that influenza virus infections induce loss of a SUMO-modified form of TRIM28 leading to a derepression of ERVs [30]. The resulting dsRNA-mediated type I IFN response support the host antiviral response, illustrating how an aberrant transcription of self-RNA derived from ERVs can be sensed as non-self by PRRs during an infection. Nevertheless, the IFN-antagonist protein NS1 produced by the wild-type influenza virus can oligomerize around dsRNAs, showing how the virus can adapt to mitigate the antiviral effects of ERVs transcription [30].

Section 2.3 has been completed to introduce ref 4 (page 4, lines 168-172)

Recently, it has been shown that interferon-γ-inducible protein 16 (IFI16), an AIM2-like receptor for DNA, is able to sense HERV-K-derived single-stranded DNA (ssDNA) in somatic tissues, although being absent in stem cells [34]. It seems likely that the immune system evolved different mechanisms to provide a better control of an aberrant transcription of these sequences in somatic tissues while allowing in stem cells where they contribute to cell identity.

A new section on immune suppression have been developed to introduce ref. 3 (page 6, lines243-260)

3.3. Suppression of the immune response

Most exogenous retroviruses have been described to suppress the host’s immune system to maintain the infection [48]. Concerning ERVs, this effect has been co-opted for the foeto-maternal tolerance, promoted by the expression of Syncitin 2, a HERV-FRD-derived Env protein, in the placenta [49]. This shared ability led to the identification of an immunosuppressive domain (ISD) in the transmembrane unit of the Env protein with multiple immunosuppressive effects (reviewed in [50]). Exposition of human PBMCs to ISD-derived peptides from different retroviruses including HIV and HERV-K increase the expression of different cytokines including IL-10, IL-6, IL-8, RANTES, MCP-1, MCP-2, TNF-α , MIP-1α , MIP-1β , MIP-3, IL-1β and Gro(α,β,γ) and decrease the expression of IL-2 and CXCL9 [51,52]. Although all studies to date converge to a direct effect on cellular immune effectors, the exact impact of Env protein’s ISD remain unclear, and no interaction with the innate immune response has been clearly highlighted. An interesting study have been performed in zebra-fish embryos, which have only innate immunity during the first month of development [53]. Embryos injected with HERV-K-containing extra-cellular vesicles derived from two colorectal adenocarcinoma cell lines treated by decitabine, showed a reduced expression of IL-1β and myeloperoxidase. This effect was however modest and need to be investigated in other models.

Reviewer 2 Report

I suggest the authors to smooth out their written English. For example in the Abstract section, the following sentence should be modified to deliver it more comprehensible to the readers: “Still, their exact role in the innate response, particularly in cancer, is still to define.” I therefore suggest modifying this sentence as: However, their exact role in the innate immune response, particularly in cancer remains to be defined.

Author Response

We would like to thank reviewer 2 for his global comments on our work. Modifications have been made consequently to smoothen the whole manuscript:

In the abstract, page 1 lines 21-22: the sentence has been replaced.

However, their exact role in the innate immune response, particularly in cancer remains to be defined.

Page 2 line 86: Slight sentence modification

Page 3 line 93: Slight sentence modification

Page 3 line 106: slight headline modification

Page 3 line 116: slight sentence break/modification

Page 3 lines 123-124: slight sentence modification

Page 3 line 132: doublestranded RNA -> dsRNA

Page 5 lines 192-193: A sentence has been added

In line with this, an impaired activity of IFN-stimulated genes is observed in cell lines deleted of these HERV sequences (36).

Page 5 paragraph 3.1: slight sentences modifications

Page 7 lines 289-291, 298-299 : slight sentences modifications

Page 7 line 328: A word has been added

Page 8 lines 342-343: slight sentences modifications